# Is Acupuncture an Ideal Adjunctive Treatment for Cancer-Related Fatigue? Comment on Choi et al. Acupuncture for Managing Cancer-Related Fatigue in Breast Cancer Patients: A Systematic Review and Meta-Analysis. *Cancers* 2022, *14*, 4419

**DOI:** 10.3390/cancers15010223

**Published:** 2022-12-30

**Authors:** Xiaoqian Hu, Beibei Feng, Jindong Xie, Xinpei Deng, Yutian Zou

**Affiliations:** 1School of Biomedical Sciences, The University of Hong Kong, 21 Sassoon Road, Hong Kong SAR, China; 2Department of Rehabilitation Medicine, The Sixth Affiliated Hospital of Sun Yat-sen University, Guangzhou 510655, China; 3State Key Laboratory of Oncology in South China, Department of Breast Oncology, Collaborative Innovation Center for Cancer Medicine, Sun Yat-sen University Cancer Center, 651 Dongfeng East Road, Guangzhou 510060, China

We read with great interest the results of the systematic review and meta-analysis conducted by Choi et al. showing the beneficial effects of acupuncture (AT) on cancer-related fatigue (CRF) in patients with breast cancer that was published in your esteemed journal (Volume 14, Issue 18) and congratulate the authors [1]. Overall, we believe the publication of this article will strengthen the benefits of AT for CRF and motivate oncologists to consider high-quality randomized controlled trials (RCTs) and reviews for validating the therapeutic effects of AT in subgroups of cancer patients classified by demographic and disease characteristics.

CRF is one of the most common and disabling symptoms for people with breast cancer [2,3]. Considering the lack of effective pharmacological strategies for the management of CRF, acupuncture, as recommended by the US National Comprehensive Cancer Network (NCCN) guidelines, has been widely adopted for the management of CRF, whereas its efficacy remains inconsistent [4]. The authors performed a meta-analysis and included data from 12 RCT studies with a total of 1084 participants. They found that AT yielded a better effect on CRF, which was measured by a continuous variable–fatigue score, when compared with both sham and usual control treatment. Given the poor quality and limited size of the evidence, the authors concluded that further research is required to reach a decisive clinical recommendation.

We agree with the meta-analysis results that treating CRF in breast cancer patients with AT remains a wait-and-watch policy; although a trend was observed in most of the included studies in favor of AT, this failed to reach significance. Despite the significant effect size in ameliorating breast-cancer-associated CRF with AT, 9 out of 12 included studies were judged with a moderate to high risk of bias, which may lead to the lack of objectivity and universality of meta-analysis conclusions. Therefore, we recommend the authors search for more recent studies which are generally of better quality in a broader range of databases, such as MEDLINE, AMED, and PsycINFO. Also, data from RCTs for patients with different cancers can be retrieved and included if diagnoses of breast cancer have been specified [5]. For those included studies, it is acknowledged that blinding patients to AT treatment might not be practical, while methods to reduce such risk of bias should be under careful consideration in future RCTs.

In addition to AT, a wide variety of complementary and alternative medical treatments have been recommended for CRF [6]. Nonpharmacologic interventions like exercise, psychoeducation, cognitive behavioral therapies, and mindfulness have produced strong evidence for their efficacy in managing CRF [2]. However, it is still unclear whether AT could yield a similar effect as these popular strategies for treating CRF in breast cancer patients. Thus, we suggest the authors perform a network meta-analysis (indirect comparisons), which allows the assessment of the relative effects of different intervention types on CRF in breast cancer patients during or after cancer treatment. Additionally, AT should not be seen as a replacement for pharmacological interventions in alleviating cancer-related symptoms, but rather an adjunct with a low level of procedure-related complications that might enhance treatment efficacy and, to some extent, allow for minimizing dosage of other drugs and related side effects. In such cases, the synergistic effect of AT and other treatments for CRF should be further considered. Although AT has a long history as a safe practice in the hands of skilled practitioners, the safety of combining AT in everyday practice with other CRF treatment still needs to be carefully considered in future study to establish applicable guidelines and encourage standardized clinical practice.

As for implications for future practice and research, despite smaller numbers, there was a clear benefit of managing CRF in breast cancer patients with AT. It is highly probable that a trend towards beneficial effects would have reached significance had the RCTs completed accrual. It is noteworthy that a broad range of uni- and multi-dimensional CRF-assessing measures were adopted in the included RCTs, which could hamper direct combination and comparison among studies. The implementation of multi-dimensional assessment is highly recommended in future research, as the beneficial effects of AT on both emotional and physical fatigue status could be thoroughly evaluated. In addition, as AT essentially regulates the vagal network and restores functionality across all systems, fMRI evidence of pre- and post-intervention in at least a percentage of participants is encouraged in future studies to further support the efficacy of AT in CRF management. Moreover, consensus on the gold-standard measure for assessing CRF is imperative. It is difficult to adequately power high-quality RCTs and reviews without universal agreement on cut-off scores and minimal clinically important differences for the selected measures.

Furthermore, it is recommended that subgroup analysis for the effect of AT for CRF in breast cancer patients based on disease stages should be performed in future RCTs and reviews. The trajectory of CRF could vary in accordance with the stage of cancer, with physical but not mental fatigue reported to be more critical for patients with advanced-stage cancer compared to cancer survivors who were off treatment [7]. While most included studies failed to compare the therapeutic effect of AT for CRF of breast cancer patients in different disease stages, further investigation is warranted to validate whether AT yields similar effects for cancer patients in early and advanced disease trajectory.

While AT emerges as a promising intervention to ameliorate CRF in patients with breast cancer, widespread implementation is still inconclusive and can be complex and resource intensive. Therefore, the wider use of AT in clinical practice for managing CRF needs to be justified by evidence supporting that potential outcome benefits outweigh the direct costs of AT from future RCTs and reviewed with high quality.

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
