# Peer review of "Is Acupuncture an Ideal Adjunctive Treatment for Cancer-Related Fatigue? Comment on Choi et al. Acupuncture for Managing Cancer-Related Fatigue in Breast Cancer Patients: A Systematic Review and Meta-Analysis. Cancers 2022, 14, 4419"

_cancers, 2022, doi:10.3390/cancers15010223_

Round 1

Reviewer 1 Report

Thank you for these comments on the future research needed for AT for the support of CRF. Whilst I agree with most of what is said, safety is one area that I do not think needs testing in the way described. AT has a long history of being a safe practice in the hands of skilled practitioners. I do not think that a lot of safety testing research is needed for the continued use of AT for CRF. This would not be a good use of future research funding.

Author Response

Responses to Reviewer 1:

Thank you for these comments on the future research needed for AT for the support of CRF. Whilst I agree with most of what is said, safety is one area that I do not think needs testing in the way described. AT has a long history of being a safe practice in the hands of skilled practitioners. I do not think that a lot of safety testing research is needed for the continued use of AT for CRF. This would not be a good use of future research funding.

Comment 1: “Thank you for these comments on the future research needed for AT for the support of CRF.”

Response: Thank you very much for the reviewer’s comments.

Comment 2: “Whilst I agree with most of what is said, safety is one area that I do not think needs testing in the way described. AT has a long history of being a safe practice in the hands of skilled practitioners. I do not think that a lot of safety testing research is needed for the continued use of AT for CRF. This would not be a good use of future research funding.”

Response: Thank you very much for pointing this out. We strongly agree that “AT has a long history of being a safe practice in the hands of skilled practitioners”, so evaluating the safety of AT for CRF should not be rank as the first priority in future study. This comment has been added in the manuscript accordingly. Since patients with cancer may be at a higher risk of developing adverse effects due to the compromised immune function and concomitant chemotherapy, they are more susceptible to infections and hemorrhage during AT treatment. Thus, using AT for CRF can be a more conducive practice if associated safety issues are recognized and well managed by skilled practitioners.

Reviewer 2 Report

Excellent and very helpful comments for future investigators and meta analyses.

I agree on all suggestions:

Widen the recruitment to all cancers.

Bias to be better managed by study design and notified on STRICTA.

The stage of the cancer and how helpful AT is at that time is also useful.

AT essentially re regulates the vagal network and restore all functionality across all systems (caused by the insult - in this case the cancer and related management). It would be good to encourage fMRI evidence of pre and post intervention in at least a percentage of the participants in future studies.

Author Response

Responses to Reviewer 2:

Excellent and very helpful comments for future investigators and meta-analyses.

I agree on all suggestions:

Widen the recruitment to all cancers.

Bias to be better managed by study design and notified on STRICTA.

The stage of the cancer and how helpful AT is at that time is also useful.

AT essentially re regulates the vagal network and restore all functionality across all systems (caused by the insult - in this case the cancer and related management). It would be good to encourage fMRI evidence of pre and post intervention in at least a percentage of the participants in future studies.

Comment 1: “Excellent and very helpful comments for future investigators and meta-analyses. I agree on all suggestions: Widen the recruitment to all cancers. Bias to be better managed by study design and notified on STRICTA. The stage of the cancer and how helpful AT is at that time is also useful.”

Response: Thank you very much for the reviewer’s comment.

Comment 2: “AT essentially re regulates the vagal network and restore all functionality across all systems (caused by the insult - in this case the cancer and related management). It would be good to encourage fMRI evidence of pre and post intervention in at least a percentage of the participants in future studies.”

Response: Thank you very much for providing such a great suggestion. We have added this point in the manuscript accordingly.
